# Patient Self-Inflicted Lung Injury—A Narrative Review of Pathophysiology, Early Recognition, and Management Options

**DOI:** 10.3390/jpm13040593

**Published:** 2023-03-28

**Authors:** Peter Sklienka, Michal Frelich, Filip Burša

**Affiliations:** 1Department of Anesthesiology and Intensive Care Medicine, University Hospital Ostrava, 17. listopadu 1790, 70800 Ostrava, Czech Republic; 2Department of Intensive Medicine, Emergency Medicine and Forensic Studies, Faculty of Medicine, University of Ostrava, Syllabova 19, 70300 Ostrava, Czech Republic; 3Institute of Physiology and Pathophysiology, Department of Intensive Care Medicine and Forensic Studies, Faculty of Medicine, University of Ostrava, Syllabova 19, 70300 Ostrava, Czech Republic

**Keywords:** patient self-inflicted lung injury, respiratory effort, work of breathing, transpulmonary driving pressure, extracorporeal membranous oxygenation

## Abstract

Patient self-inflicted lung injury (P-SILI) is a life-threatening condition arising from excessive respiratory effort and work of breathing in patients with lung injury. The pathophysiology of P-SILI involves factors related to the underlying lung pathology and vigorous respiratory effort. P-SILI might develop both during spontaneous breathing and mechanical ventilation with preserved spontaneous respiratory activity. In spontaneously breathing patients, clinical signs of increased work of breathing and scales developed for early detection of potentially harmful effort might help clinicians prevent unnecessary intubation, while, on the contrary, identifying patients who would benefit from early intubation. In mechanically ventilated patients, several simple non-invasive methods for assessing the inspiratory effort exerted by the respiratory muscles were correlated with respiratory muscle pressure. In patients with signs of injurious respiratory effort, therapy aimed to minimize this problem has been demonstrated to prevent aggravation of lung injury and, therefore, improve the outcome of such patients. In this narrative review, we accumulated the current information on pathophysiology and early detection of vigorous respiratory effort. In addition, we proposed a simple algorithm for prevention and treatment of P-SILI that is easily applicable in clinical practice.

## 1. Introduction

Continuous oxygen supply and carbon dioxide elimination are the principal physiologic functions of the respiratory system. This gas exchange requires a large epithelial surface in direct contact with both the external (air) and internal (blood flow through the pulmonary circulation) environments. Lungs and bronchial epithelium also serve as first-line immune defense organs, including innate and adaptive immune cells capable of inducing the local and systemic immune response. The energy needed to generate a pressure gradient between alveoli and atmospheric pressure to pass the inspired gas through airways and to distend lungs could be produced by respiratory muscles, by a ventilator, or by a combination of these two sources. The term “biotrauma” was introduced to describe the translation of energy applied to the respiratory system into biochemical signals generating inflammatory response and subsequent dysfunction of distal organs [1]. The application of excessive mechanical energy on the lung tissue during breathing is translated into biological signals producing systemic inflammation. This explains why most patients dying with acute respiratory distress syndrome (ARDS) die from multiple organ dysfunction syndrome [2]. During mechanical ventilation (MV), the ventilator provides the work of breathing; this can, however, result in lung injury—so-called ventilator-induced lung injury (VILI). The identification of the potentially preventable factors of VILI development was a logical next step in the research on VILI. Lung-protective ventilatory strategies based on low tidal volumes, reduced pressures delivered by the ventilator (tidal volume (V_t_) ≤ 6 mL/kg of predicted body weight (PBW); plateau pressure ≤ 30 cmH_2_O; driving pressure (ΔPaw)  ≤  15 cmH_2_O), and optimization of positive end-expiratory pressure (PEEP) were associated with reduced mortality in patients with ARDS [3]. Finally, the concept of mechanical power was introduced into the clinical practice for calculating the energy applied to respiratory tissue during mechanical ventilation [4]. 

In a spontaneously breathing subject, the respiratory effort is generated by inspiratory muscles and the lung tissue is, therefore, exposed to the same physical forces that cause lung injury accompanied by VILI. If applied to damaged lung tissue, this can also lead to further lung injury, and the fact that this is caused by spontaneous breathing is reflected in the term “patient self-inflicted lung injury” [5]. It is important to mention that P-SILI might develop in patients on mechanical ventilation with preserved spontaneous effort as well as in patients breathing without mechanical ventilator support. Because the exact measurement of driving pressure and tidal volume in a patient breathing spontaneously has significant limitations, clinicians must be aware of the early signs of excessive work of breathing and respiratory effort. 

In this narrative review, we accumulated the current information on pathophysiology and, in particular, early detection of P-SILI and options for its treatment. The knowledge provided in this paper may help clinicians in the early recognition of patients at risk and, in effect, in applying early interventions to prevent the development of irreversible lung injury. 

## 2. Experimental Evidence of P-SILI

The early evidence of a detrimental effect of excessive inspiratory effort was described in the experimental study on pharmacologically induced hyperventilation (central acidosis induced by sodium salicylate injection into the cisterna magna) in spontaneously breathing animals with healthy lungs. Hyperventilation lasting for a prolonged period induced subsequent hypoxemia, deterioration of lung mechanics (decrease of compliance), and macroscopic and histological features consistent with ARDS. These lesions were not detected in animals in the control arm, where objects were sedated, paralyzed, and mechanically ventilated with average tidal volumes immediately after the application of sodium salicylate injection into the cisterna magna [6]. Yoshida et al. reported the deleterious effect of excessive respiratory effort in the lavage-injured lung in animals with the inserted esophageal balloon and monitored esophageal pressure (P_es_). Animals with strong spontaneous effort (defined by a greater value of the negative pleural pressure and increased esophageal pressure switch) developed a significant worsening of the lung injury as evaluated by the dynamic compliance, neutrophil count in the bronchoalveolar fluid, and histological lung injury score [7]. The same authors defined the role of lung injury severity in the development and progression of P-SILI in animals with severe lung injury induced by repeated lung lavage. Histological lung injury scores were higher in the subjects with initial severe lung injury who were left to breathe spontaneously compared to those who were put on mechanical ventilation without spontaneous effort. However, animals with initial mild lung injury who were left to breathe spontaneously showed the best results of all groups, thus highlighting the need for individualization and meticulous evaluation of the suitability of spontaneous breathing for a particular subject with lung injury [8]. An excessive spontaneous effort with high regional strain was also shown to be the cause of the increased expression of genes involved in inflammation, coagulation, and apoptosis [4]. These data from experimental studies, therefore, provide evidence for the link between excessive respiratory effort and the subsequent development of self-inflicted lung injury. Some authors advocate the use of the term “effort-induced lung injury” instead of P-SILI [9]. 

## 3. Pathophysiology of P-SILI 

The pathophysiology of P-SILI is complex and involves factors related to the underlying lung pathology and respiratory mechanics. From the physiological point of view, three distinct factors determine the respiratory mechanics: (i) respiratory drive (i.e., neural output to respiratory muscles), (ii) respiratory effort (i.e., the activity of respiratory muscles), and (iii) breathing pattern (i.e., the mechanics of respiration).

**Respiratory drive.** The respiratory drive is defined as the intensity of the neural output of the respiratory centers, which determines the effort of the respiratory muscles. The respiratory drive is determined by signals from central chemoreceptors, peripheral chemoreceptors, stretch receptors from the thoracic wall and lung tissue, irritant receptors of the airway epithelium, and, finally, cortical and emotional feedback. In a pathological condition, the tissue of the respiratory system produces vital positive feedback, further increasing the respiratory drive through hypoxemia, hypercarbia, and vagal C-fibers activation by inflammation and congestion [10,11].

**Inappropriate lung stress and strain.** Lung stress is the pressure distending the lung and chest wall. Numerically, the stress applied to the lung parenchyma is represented by the transpulmonary pressure (P_L_). Strain is the change in lung volume above the end-expiratory lung volume in response to an applied stress (i.e., tidal volume). Inappropriate stress and strain are both related to the development of VILI and P-SILI. In injured lungs with reduced compliance, higher transpulmonary pressure and work of breathing are needed to provide an appropriate tidal volume and minute ventilation. Moreover, the distribution of stress and strain becomes significantly non-homogeneous under pathologic conditions, causing further deleterious regional amplification of stress and strain between regions with different mechanical properties [12,13,14].

**Pendelluft** is characterized by redistributing the tidal volume from the non-dependent to the dependent lung regions due to the higher negative pressure in the dependent areas. Moreover, in patients with substantial expiratory effort, the de-recruitment of dependent lung regions during exhalation may increase the pendelluft during the subsequent inspiration [15,16]. Intraparenchymal air shift causes significant regional over-distension of the dependent regions. It increases the work of breathing irrespective of the tidal volume, and the frequency and magnitude of pendelluft correlate with increased blood concentrations of inflammatory biomarkers [17].

**Lung edema.** Vigorous inspiratory effort and excessive intrathoracic negative pressure result in an increase in venous return. The subsequent elevation of left ventricular end-diastolic and pulmonary capillary pressures creates a high trans-capillary pressure gradient. In acute lung injury, capillary permeability is increased due to the dysfunction of the endothelial cell layer. This combination of elevated trans-capillary pressure gradient and increased capillary permeability facilitate fluid leakage from the pulmonary capillaries into the interstitial and alveolar space [18]. Lung edema, in turn, increases respiratory effort because of worsening hypoxemia and hypercarbia and due to direct stimulation of vagal C-fibers [10].

**Diaphragmatic injury.** Atrophy due to prolonged disuse in controlled mechanical ventilation was considered the primary mechanism of ventilation-induced diaphragmatic injury (VIDI). However, concentric load-induced injury caused by excessive muscle fibers contraction in patients with high respiratory effort (both inspiratory and expiratory) may also contribute to VIDI development. Because the activity and function of the diaphragm are the main factors influencing lung volume and pressure changes during spontaneous breathing, avoiding insufficient and/or excessive respiratory effort became the cornerstones of the so-called lung- and diaphragm-protective ventilation strategies [19,20].

**Patient–ventilator dyssynchrony** during mechanical ventilation is defined as a situation when the parameters of breath delivered by the ventilator do not match the patient’s respiratory drive and effort. Dyssynchronies are of clinical relevance because the number and intensity of patient–ventilator dyssynchronies are related to parameters of poor outcome, such as increased mortality or prolonged ventilation due to slower weaning [21,22,23]. 

Injured lungs are susceptible to subsequent insults. Paradoxically, respiratory insufficiency increases the respiratory drive through the central and peripheral receptor back loop and the direct stimulation of pulmonary vagal C-fibers by inflammation and edema [10,11]. The excessive respiratory drive may overwhelm lung-protective reflexes, which, in turn, leads to the breathing pattern causing self-inflicted lung injury and exposes the lungs to the risk of aggravating the pre-existing lesions—the vicious cycle of P-SILI [5] [Figure 1]. 

The COVID-19 outbreak triggered intensive research in the field of self-inflicted lung injury. In the early phase of the pandemic, several “phenotypes” of COVID-19-related ARDS were believed to exist; however, with a growing knowledge of COVID-19-related ARDS, it became apparent that the differences observed in lung pathology and clinical presentation were rather manifestations of disease development in time than different phenotypes [24,25]. Moreover, these differences may have also arisen due to a combination of disease-related conditions and respiratory effort-related injury [26]. COVID-19-related ARDS was specific because patients frequently tolerated severe hypoxemia (“silent hypoxemia”) due to the impaired function of central and peripheral chemoreceptors [27,28,29]. Patients, therefore, often developed excessive respiratory effort inducing severe P-SILI for an extended period. In most severe cases, the pressure gradient applied to the injured lung during spontaneous breathing was capable of eventually causing structural disruption of the pulmonary tissue, resulting in the most severe forms of P-SILI associated with increased mortality—spontaneous pneumothorax, pneumomediastinum, and pneumopericardium [30,31]. In retrospective studies describing the incidence and clinical features of spontaneous pneumothorax and pneumomediastinum in patients with COVID-19-related pneumonia, a significant portion of patients presented pneumothorax or pneumomediastinum at admission or during the hospital stay before initiation of any form of invasive ventilation, suggesting the association between vigorous spontaneous ventilation and progression of lung injury [32,33]. 

## 4. Identification of Risk Factors for the P-SILI Development 

Early identification of markers of vigorous effort is of fundamental importance for treating patients at high risk of P-SILI. Below, we will focus on characteristics that can be monitored at the bedside with standard equipment and minimal invasiveness. 

**Work of breathing scale.** Modern intensive medicine is based on precise monitoring and therapeutic algorithms arising from the results of clinical trials. However, physiology, pathophysiology, and clinical examination remain the cornerstones of the best clinical practice. In an excellent review, Tobin focused on simple clinical signs of increased respiratory effort and work of breathing, such as nasal flaring, sternomastoid muscle phasic contraction, tracheal tug (downward motion of the trachea with inspiratory effort), suprasternal fossa excavation during inspiration, and abdominal muscle use [34]. Nasal flaring, sternocleidomastoid muscle activity, abdominal muscle use, and respiratory rate are also included in the so-called work of breathing (WOB) scale, a simple tool for non-invasive respiratory effort monitoring and prediction of need for intubation in spontaneously breathing patients. Apigo et al. have shown in a small sample study that the use of respiratory accessory muscles resulting in WOB > 4 represents a relevant signal of subsequent high-flow nasal (HFNC) therapy failure [35]. 

**The ROX index** is a clinical tool defined as ROX index = (SpO_2_/FiO_2_)/RR, where SpO_2_—peripheral oxygen saturation; FiO_2_—fraction of inspired oxygen; and RR—respiratory rate. The ROX index was proposed as a tool for predicting the failure of HFNC in pneumonia patients with hypoxemic respiratory failure [36]. The validity of the ROX index was repeatedly confirmed in clinical studies, and its correlation with HFNC failure was confirmed in the meta-analyses of studies in patients with COVID-19-related respiratory failure [37,38]. 

**The HACOR** score, taking into account the hemodynamic response (heart rate; HR), acid-base status (pH value), consciousness (Glasgow Coma Scale; GCS), arterial oxygenation (PaO2), and respiratory rate (RR), was designed to predict a non-invasive ventilation failure [39]. In moderate to severe ARDS patients on noninvasive ventilation (NIV) as a first-line therapy, the reduction of HACOR score after 1–2 h of NIV identified the patients who responded well to NIV, indicating a lower risk of need for intubation as well as of mortality [40].

Although the WOB scale, ROX index, and HACOR score are not tools for direct measurement of the respiratory effort, such combinations of the parameters of gas exchange and respiratory mechanics might suggest an increased risk of P-SILI in spontaneously breathing patients or in patients on NIV. Moreover, analyzing the temporal trend of the WOB scale, ROX index, and/or HACOR score in a patient might help clinicians prevent unnecessary intubation and identify patients who, on the contrary, would benefit from early intubation before severe P-SILI develops.

**Esophageal and transpulmonary pressure.** The inspiratory effort exerted by the respiratory muscles is a crucial physiological mechanism contributing to P-SILI development in patients with spontaneous breathing activity. The esophageal pressure monitoring is considered clinical gold standard for the measurement of the pressure generated by the respiratory muscles (i.e., the respiratory muscle pressure, P_mus_). The changes in P_es_ (esophageal pressure swings; ΔP_es_) mirror the changes in pleural pressure, and the difference between the airway pressure (P_aw_) and P_es_ represents the transpulmonary driving pressure [41,42]. However, routine bedside P_es_ monitoring is limited by its invasiveness (esophageal balloon manometry) and potential confounding factors (misplacement of the probe, technical complications, risk of esophageal pressure ulcers, etc.). Therefore, alternative P_mus_ assessment tools were developed and validated in experimental and clinical trials. Among these methods, the monitoring of nasal pressure swings, airway occlusion pressure at 100 ms, expiratory occlusion pressure, flow index, the electrical activity of the diaphragm, ventilator waveform analysis, and electrical impedance tomography (EIT) represent simple, non-invasive or minimally invasive tools that are easy to use in daily practice. 

**Nasal pressure swings.** Tonelli et al. evaluated the correlation between ΔPes and nasal pressure swings (ΔP_nos_) in 61 consecutive spontaneously breathing patients with acute respiratory failure. One nostril was hermetically closed by the monitoring system (“nasal plug”), and HFNC was placed into the remaining patent nostril. Patients then breathed with their mouths closed, and nasal pressure swings were monitored. Interestingly, the insertion of the nasal plug did not affect inspiratory effort and respiratory rate. ΔP_es_ and ΔP_nos_ strongly correlated on admission and 24 h apart, suggesting the possibility of estimating esophageal pressure swings from minimal invasive nasal pressure swings [43]. 

**Airway occlusion pressure at 100 ms (P_0_._1_).** P_0_._1_ is defined as the negative airway pressure developed during a brief (100 ms after the onset of the inspiration) airway occlusion. Levels of P_0_._1_ less than 1.0 cm H_2_O suggest an inappropriate respiratory effort and values greater than 3.5 cm H_2_O suggest a vigorous respiratory effort [44]. Recently, pooled data from clinical studies showed P_0_._1_ to be a reliable bedside tool for monitoring the respiratory drive and detecting potentially injurious inspiratory effort [45]. Thanks to its non-invasiveness and simple monitoring, P_0_._1_ has become a routine parameter automatically monitored by modern ventilators.

**Expiratory occlusion pressure (ΔP_occ_)** is defined as the total swing in airway pressure generated by respiratory muscle effort on assisted ventilation when the airway is occluded. Bertoni et al. studied the relationship between ΔP_occ_ and parameters derived from esophageal manometry (P_mus_) and dynamic transpulmonary driving pressure (ΔP_L,dyn_) in sixteen patients ventilated in pressure support mode. Excessive respiratory effort (P_mus_ > 10 cm H_2_O) and transpulmonary driving pressure (ΔP_L,dyn_ > 15 cm H_2_O) were quite common. The authors found a significant correlation between ΔP_occ_ and both of P_mus_ and ΔP_L,dyn_, and concluded that measuring ΔP_occ_ allows non-invasive detection of elevated respiratory muscle pressure and transpulmonary driving pressure [46]. Similar results were reported by Roesthuis et al. in a study on mechanically ventilated patients with COVID-19-related respiratory failure. Although only a weak correlation was generally found between the measured and computed ΔP_L_ and P_mus_, excessive effort (ΔP_L_ > 20 cm H_2_O and P_mus_ > 15 cm H_2_O) was detected with high sensitivity and specificity [47].

**Flow index** is a novel non-invasive method based on the analysis of the flow–time curve in patients on pressure-support ventilation (PSV) to estimate the inspiratory effort. In these patients, the concavity of the inspiratory flow-time waveform recorded during pressure support ventilation reflects the inspiratory effort because the effort-induced pressure difference between the airway opening and the alveoli determines the intensity of the inspiratory flow. Flow index can be calculated using relatively simple software. In small-sample clinical trials, the calculation of inspiratory effort by flow index agreed with esophageal-pressure-based methods [48,49].

**Electrical Activity of the Diaphragm.** Electrical activity of the diaphragm (EAdi) provides important information about the crural diaphragm. The P_mus_/EAdi (PEI) index characterizes the pressure generated by the respiratory muscles in relation to the electrical activity of the diaphragm, which facilitates the discrimination between the respiratory drive (electrical activity) and the respiratory effort (generated pressure) [50,51]. The limitation of EAdi measurement lies in its invasiveness, but small-sample clinical studies confirmed a significant correlation between EAdi and the diaphragmatic electrical activity measured by surface electromyography [52,53]. Surface diaphragmatic electromyography may, therefore, serve as a non-invasive alternative for inspiratory effort monitoring.

**Ventilator waveforms analysis** provides crucial information on patient-ventilator physiology and interaction. Visual analysis of flow/time and pressure/time curves alongside clinical examination provide information on patient–ventilator interaction and dyssynchrony. 

**Electrical impedance tomography** (EIT) is a non-invasive, bedside functional imaging modality allowing breath-to-breath visualization of lung ventilation. During spontaneous breathing, EIT can detect the non-homogeneous distribution of lung ventilation and the pendelluft phenomenon [54].

In Figure 2, noninvasive or minimally invasive bedside tools for detection of potentially injurious respiratory effort are summarized.

## 5. Prevention and Treatment of P-SILI 

**Awake-prone position.** The effect of early prolonged prone position (PP) was evaluated in 81 awake patients with acute COVID-19-associated respiratory failure requiring noninvasive ventilatory support. Prone position was associated with fewer incidences of NIV failure (defined as the need for intubation) and mortality. Moreover, physiological effects arising from PP were observed—lung tissue aeration (defined by lung ultrasound score), paO_2_/FiO_2_, respiratory rate, and plasma levels of inflammatory biomarkers significantly improved in patients who underwent PP [55]. These effects of pronation strongly suggest that the prone position is associated with a reduced risk of self-inflicted lung injury [56]. The favorable effect of awake prone positioning on paO_2_/FiO_2_ and respiratory rate in patients with spontaneous breathing was confirmed in recent meta-analyses [57,58]. 

**Sedation** reduces the respiratory drive and may protect the lung from further aggravation of P-SILI. However, long-term sedation is accompanied with increased risk of ventilator-associated pneumonia, muscle weakness (including diaphragm), and subsequent delirium. During the COVID-19 pandemic, high doses of intravenous sedatives and opioids were reported to achieve sedation and suppress patient–ventilator dyssynchronies [59]. It is noteworthy that sedation scales used in intensive medicine are markers of arousal but do not correspond to the intensity of respiratory drive. In patients on mechanical ventilation, respiratory drive monitored by P_0_._1_ did not correlate with the Richmond Agitation–Sedation Scale [60]. The approach of personalized sedation aiming at maintaining safe levels of dyssynchrony and patient effort while simultaneously allowing spontaneous ventilation is termed “lung-protective sedation” [61].

In mechanically ventilated patients with ARDS who were (according to the mechanical parameters of the respiratory system) considered eligible for partial support ventilation, a reduction in intravenous sedation was accompanied by excessive respiratory effort in as much as 80% of cases [62]. Therefore, alternative sedation strategies, such as partial neuromuscular blockade or inhalation sedation, should be considered. 

**Inhalation sedation** allows spontaneous breathing in patients on mechanical ventilation while in deep sedation. Landoni et al.’s meta-analysis found a reduced need for intravenous sedatives and opioids when using inhalation sedation [58]. Alongside sedation, inhaled anesthetics may provide other clinical benefits—bronchodilatation, reduced inflammation, improved gas exchange, and facilitated awakening [63,64].

**Partial neuromuscular blockade** is a pharmacological strategy to treat high tidal volumes and negative pressure swings caused by excessive respiratory effort in patients on partial ventilatory support. This is achieved by applying neuromuscular blocking agents in low doses to maintain diaphragm activity and spontaneous ventilation [65,66]. 

**Early protective mechanical ventilation.** Although evidence from experimental and clinical studies suggests that P-SILI is a severe and life-threatening clinical condition, intensivists have no consensus on when to intubate a patient with excessive respiratory effort. In animal studies with artificially induced lung injury, early use of mechanical ventilation eventually led to less severe lung injury than leaving them to ventilate spontaneously with vigorous effort, and mechanical ventilation prevented self-inflicted lung injury if applied early but not when applied late [6,7,67,68]. Some authors, therefore, advocate that in the context of high respiratory drive and vigorous respiratory effort, mechanical ventilation with lung-protective parameters might be considered a protective therapy to minimize P-SILI [5,69].

**Spontaneous breathing during mechanical ventilation** in patients with severe lung injury remains subject to an ongoing debate amongst intensivists. Active insufflation of the lung by inspiratory muscle activity improves gas exchange and lung aeration, mainly in dependent regions, optimizes ventilation–perfusion matching, and prevents the loss of end-expiratory lung volume (i.e., of the functional residual capacity) [70,71]. Moreover, spontaneous breathing during mechanical ventilation promotes the recovery of the diaphragmatic thickness and can partially reverse respiratory muscle atrophy. However, it is still necessary to consider the prevention of excessive effort that can lead to P-SILI and consequent diaphragmatic myoinjury. In experimental study, spontaneous breathing was beneficial in animals with mild lung injury; however, in animals with severe lung injury, spontaneous breathing was associated with a significant increase in atelectasis with cyclic collapse, higher plateau pressure, and more excessive spontaneous breathing effort, resulting in the highest transpulmonary pressure and the highest driving pressure. On the other hand, muscle paralysis in severe lung injury resulted in better oxygenation and less histological lung injury [8]. Currently, the data from clinical trials are insufficient to make evidence-based recommendations and guidelines on the strategy of spontaneous ventilation in mechanically ventilated subjects in the acute phase of ARDS. According to the expert panel opinion, ventilation with a pressure mode allowing spontaneous ventilation can be used when ensuring that the tidal volume generated is close to 6 mL/kg predicted body weight (PBW) and does not exceed 8 mL/kg PBW [72,73].

**Extracorporeal membrane oxygenation (ECMO).** It is essential to mention that the beneficial effects of ECMO rather lie in facilitating lung rest and healing than in acting as a modality of lung treatment per se. Extracorporeal oxygenation and elimination of carbon dioxide could significantly reduce the patient’s respiratory drive and, therefore, reduce the risk of excessive respiratory effort in spontaneously breathing patients [74,75]. An “awake ECMO approach” is based on the implantation of venovenous ECMO in spontaneously breathing patients to avoid undesirable effects associated with mechanical ventilation and continuous deep sedation. Avoiding mechanical ventilation could be advantageous, especially in patients at high risk of pneumothorax or pneumomediastinum [76,77,78].

Hypoxemia and hypercarbia are the main reasons for instituting mechanical ventilation in patients with acute respiratory failure. During the COVID-19 pandemic, large numbers of COVID-19 patients presented with severe hypoxemia and hypercarbia but were fully conscious and without subjectively perceived severe discomfort. Lung visualization by computed tomography or ultrasound verified severe structural injury of the lung parenchyma, and the respiratory mechanics of these patients was characterized by vigorous respiratory effort. A significant portion of patients with COVID-19-related respiratory failure finally had to be put on mechanical ventilation or ECMO, and the mortality of these patients was in excess. Therefore, it is important to recognize situations when spontaneous respiration with vigorous effort becomes dangerous, further aggravating preexisting lung injury. 

Preventing P-SILI in mechanically ventilated patients with preserved spontaneous breathing activity is another important issue. Dianti et al. performed an elegant study on the strategy for minimizing the risk of P-SILI in 30 mechanically ventilated patients. The approach used to achieve lung- and diaphragm-protective targets (−3 to −8 cm H_2_O; dynamic transpulmonary driving pressure 15 cm H_2_O) was based on (i) successive steps of titration of sedation, (ii) adjustment of ventilation parameters (including the titration of the PEEP value to the maximum dynamic compliance) and sweep gas flow (in patients on ECMO), and (iii) the application of partial neuromuscular blockade. Initially, only six patients met the goals of lung- and diaphragm-protective targets, but following the study protocol, these goals were achieved in all monitored patients [62]. P-SILI prevention should be, therefore, viewed as a complex approach combining different therapeutic modalities according to respiratory mechanics and gas exchange parameters. A proposal of therapeutic algorithm in prevention of P-SILI is presented in Figure 3. 

The COVID-19 pandemic provided insight into the field of self-inflicted lung injury, but many relevant topics remain unsolved. Further research should be focused on identifying criteria strictly associated with injurious respiratory effort, defining the strategies balancing the benefits and risks between early and delayed intubation, and interventions minimizing the disadvantages arising from vigorous effort in patients on mechanical ventilation. 

## 6. Conclusions

Self-inflicted lung injury represents a life-threatening complication in patients with acute respiratory failure. During vigorous spontaneous breathing, lung tissue is exposed to the physical forces producing inappropriate stress, strain, pendelluft, and lung congestion, which induces histopathological and clinical features that further aggravate lung injury. Clinicians must be aware of the early signs of excessive work of breathing and respiratory effort. Incorporating clinical signs of the increased work of breathing alongside with WOB scale, ROX index, or HACOR score in spontaneously breathing patients may help clinicians prevent unnecessary intubation while also identifying patients who would benefit from early intubation before severe P-SILI develops. During mechanical ventilation, parameters estimating P_mus_ may suggest inappropriate respiratory effort. Prevention and treatment of P-SILI are multimodal and require detailed evaluation of respiratory mechanics and mechanisms causing the excessive effort.

## Figures and Tables

**Figure 1 jpm-13-00593-f001:**
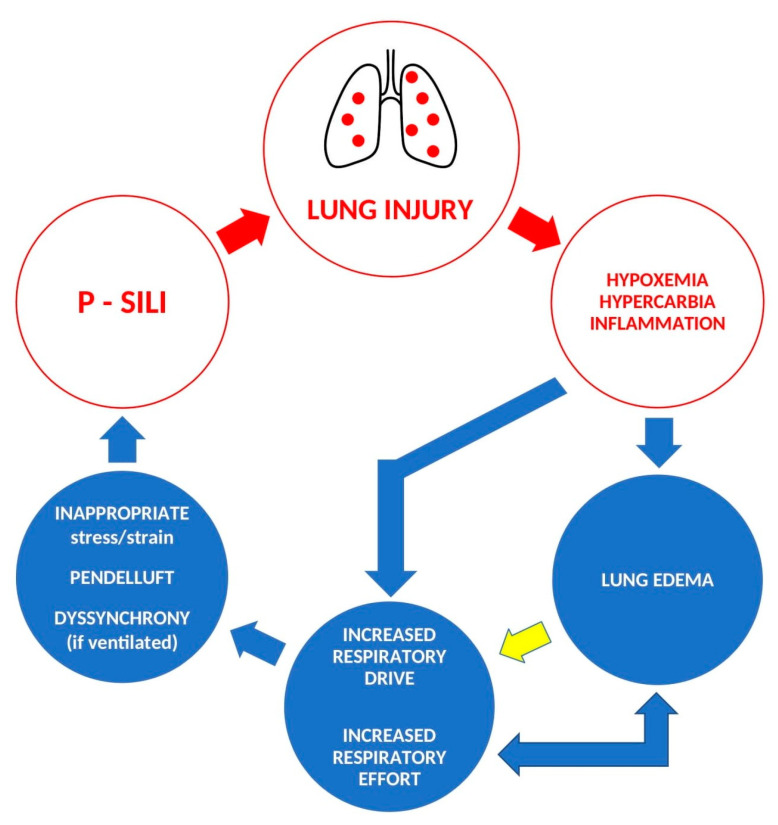
The pathophysiology of P-SILI—a “vicious circle” of self-aggravating lung injury; yellow arrow—vagal signalization (according to [5,10,18]).

**Figure 2 jpm-13-00593-f002:**
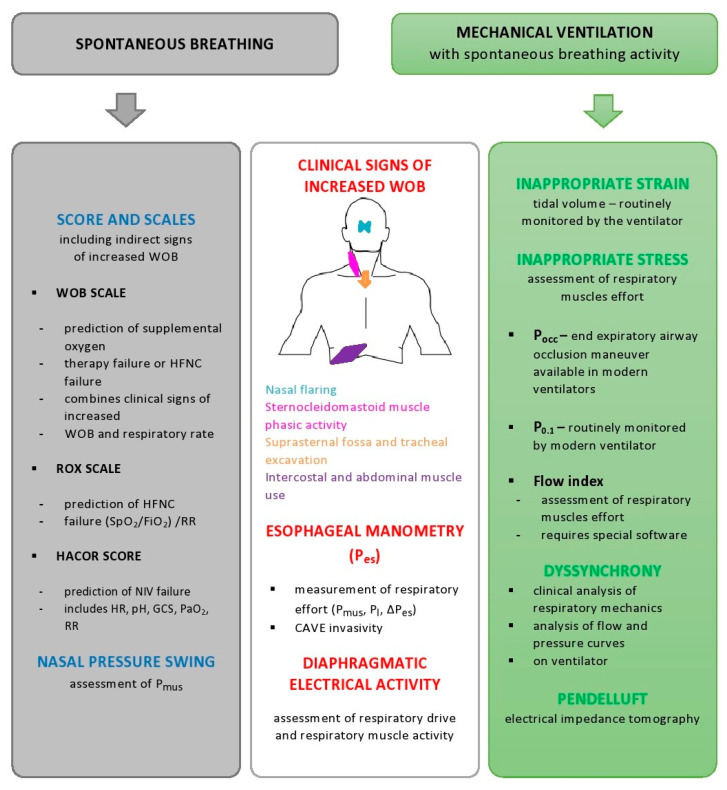
Noninvasive or minimally invasive bedside tools for detection of potentially injurious respiratory effort. Blue square—methods of indirect detection of vigorous spontaneous effort in patients breathing spontaneously or on noninvasive ventilation. White square—methods applicable both during spontaneous breathing and mechanical ventilation. Green square—methods applicable under mechanical ventilation. Abbreviations: WOB—work of breathing; HFNC—high-flow nasal cannula; NIV—noninvasive ventilation; SpO_2_—peripheral oxygen saturation; FiO2—fraction of inspired oxygen; RR—respiratory rate; HR—heart rate; GCS—Glasgow Coma Scale; paO_2_—arterial partial pressure of oxygen; P_es_—esophageal pressure; ΔP_es_—esophageal pressure swings; P_mus_—respiratory muscle pressure; P_L_—transpulmonary pressure; Vt—tidal volume; IBW—ideal body weight; P_0_._1_—airway occlusion pressure at 100 ms; P_occ_—expiratory occlusion pressure.

**Figure 3 jpm-13-00593-f003:**
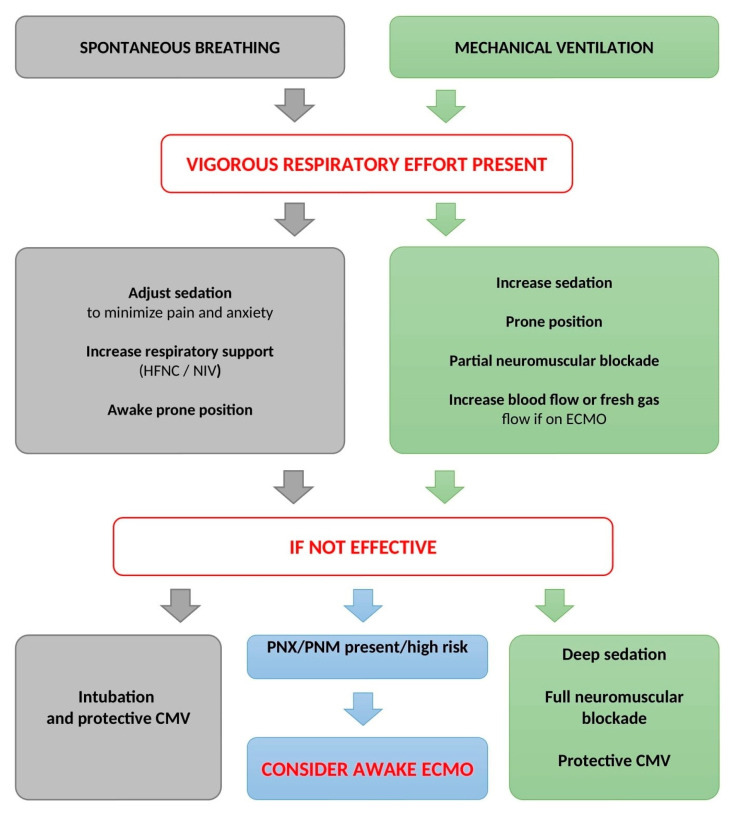
Proposal of an algorithm for P-SILI prevention and treatment. CMV—controlled mechanical ventilation; HFNC—high-flow nasal cannula; NIV—noninvasive ventilation; PNX—pneumothorax; PNM—pneumomediastinum; ECMO—extracorporeal membranous oxygenation.

## Data Availability

Not applicable.

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
