# Peer review of "Patient Self-Inflicted Lung Injury—A Narrative Review of Pathophysiology, Early Recognition, and Management Options"

_jpm, 2023, doi:10.3390/jpm13040593_

Round 1

Reviewer 1 Report

The review by Sklienk and et al deals with a very interesting topic, the Patient-Self nflicted Lung injury. The auhors present the topic starting from the experimental evidence supporting P-SILI and then deal with the pathophysiological mechanisms and discuss the risk factors and how they could be monitored to avoid the injury. Lastly, they propose methods to avoid or treat patients wih P-SILI.

Although this is an interesting paper, there are some major issues.

1. there is a lack of reference to support what is presented, throughout the manuscript. The introduction has no reference at all, and so are the paragraphs concerning stress and strain, sentence "the covid-19 outbreak-different phenotypes" in the last paragraph of the third section, the paragraph concerning esophageal and transpulmonary pressure, partial neuromuscular blockade, spontaneous breathing during MV, ECMO,

2. IN THE LAST PARAGRAPH of the third section there are some interesting ideas, but the references used are reviewes. are there any clinical data to support them?

3. diaphragmatic injury is a form of patient self inflicted injury not concerning the lungs, therefore it could be presented separately

4. the information in Figure 2 are not well presented.

Minor comments

1. prevention and treatment, there is a typo- mistake

2. What does LDP mean?

Author Response

Dear reviewer,

We thank you for your valuable comments on our article „Patient self-inflicted lung injury – a narrative review of pathophysiology, early recognition, and management options.“. We have accepted most of your comments on the article because we are convinced this will increase its quality.

According to your specific comments:

Ad 1 and 2 – we have added a few relevant references regarding every mentioned topic

Ad 3 – we focused on the so-called concentric load-induced injury from the excessive respiratory effort. This type of dysfunction could be accompanied by lung injury (references added)

Ad 4 – corrected

Ad minor comments – corrected

Yours sincerely

Peter Sklienka

Reviewer 2 Report

Very interesting article 

comments:

1. the title isn't in accordance with the next; treatment of P-SILI? there is no reference to this; rather prevention

2. line 48: more explicit on protective mechanical ventilation (parameters value)

3. line 162: refer to Gattinioni who described ARDS phenotypes 

4. line 178:  precise in the text if the technique is for patients breathing spontaneously or on mechanical ventilation

5. line 309: mention also the deleterious effects of prolonged sedation

6. not enough references for such an important matter

Author Response

Dear reviewer,

We thank you for your valuable comments on our article „Patient self-inflicted lung injury – a narrative review of pathophysiology, early recognition, and management options.“. We have accepted most of your comments on the article because we are convinced this will increase its quality.

According to your specific comments:

Ad 1 – tittle was adjusted

Ad 2 – parameters value added

Ad 3 – references added

Ad 4 – added

Ad 5 – added bellow

Ad 6 – further important references were added

Yours sincerely

Peter Sklienka

Round 2

Reviewer 1 Report

Dear Authors

The manuscript that is re-submitted has been greatly improved, and the authors should be congratulated for this. Yet there are some points that need further clarification. 

1. the first three sentences in the introducion could be omited as the meaning is too general and does not add any knowledge to the specific issue of PSILI

2. PAGE 5 last sentence of the 3rd section is just a hypothesis and should be de-empasized. the disease itself may account for the pneumomediastenum.

3. Figure 2. CAVE invasivity: this is not explained

4. Diaphragmatic ultrasonography could be presented as a means to measure the respiratory effort

5. section 5 "These effects of pronation- injury" this is not a straightforward association for PSILI  reduction.

6. section 5 third paragraph the word care should be placed between intensive medicine

7. the outcome of ref 62 should be presented

8. figure 3 "increase respiratory support HFNC, NIV" is there a limit to the increase? the transpulmonary pressure should be considered when increasing the support in this situation

Reviewer 2 Report

No other suggestion